

# Association of urban forest landscape characteristics with biomass and soil carbon stocks in Harbin City, Northeastern China

Hailiang Lv[1,2], Wenjie Wang[1,2], Xingyuan He[1,3], Chenhui Wei[1], Lu Xiao[1], Bo Zhang[2] and Wei Zhou[2]

[1] Northeast Institute of Geography and Agricultural Ecology, Chinese Academy of Sciences, Changchun, The People's Republic of China
[2] Northeast Forestry University, Harbin, The People's Republic of China
[3] Univeristy of Chinese Academy of Sciences, Beijing, The People's Republic of China

Corresponding author
Wenjie Wang, wangwenjie@iga.ac.cn, wwj225@nefu.edu.cn

## ABSTRACT

**Background**. Urban forests help in mitigating carbon emissions; however, their associations with landscape patterns are unclear. Understanding the associations would help us to evaluate urban forest ecological services and favor urban forest management via landscape regulations. We used Harbin, capital city of the northernmost province in China, as an example and hypothesized that the urban forests had different landscape metrics among different forest types, administrative districts, and urban–rural gradients, and these differences were closely associated with forest carbon sequestration in the biomass and soils.

**Methods**. We extracted the urban forest tree coverage area on the basis of 2 GF-1 remote sensing images and object-oriented based classification method. The analysis of forest landscape patterns and estimation of carbon storage were based on tree coverage data and 199 plots. We also examined the relationships between forest landscape metrics and carbon storage on the basis of forest types, administrative districts, ring roads, and history of urban settlements by using statistical methods.

**Results**. The small patches covering an area of less than 0.5 ha accounted for 72.6% of all patches (average patch size, 0.31 ha). The mean patch size (AREA_MN) and largest patch index (LPI) were the highest in the landscape and relaxation forest and Songbei District. The landscape shape index (LSI) and number of patches linearly decreased along rural-urban gradients ($p < 0.05$). The tree biomass carbon storage varied from less than 10 thousand tons in the urban center (first ring road region and 100-year regions) to more than 100 thousand tons in the rural regions (fourth ring road and newly urbanized regions). In the same urban–rural gradients, soil carbon storage varied from less than five thousand tons in the urban centers to 73–103 thousand tons in the rural regions. The association analysis indicated that the total forest area was the key factor that regulates total carbon storage in trees and soils. However, in the case of carbon density (ton ha$^{-1}$), AREA_MN was strongly associated with tree biomass carbon, and soil carbon density was negatively related to LSI ($p < 0.01$) and AREA_MN ($p < 0.05$), but positively related to LPI ($p < 0.05$).

**Discussion**. The urban forests were more fragmented in Harbin than in other provincial cities in Northeastern China, as shown by the smaller patch size, more complex patch

shape, and larger patch density. The decrease in LSI along the rural-urban gradients may contribute to the forest carbon sequestrations in downtown regions, particularly underground soil carbon accumulation, and the increasing patch size may benefit tree carbon sequestration. Our findings help us to understand how forest landscape metrics are associated with carbon storage function. These findings related to urban forest design may maximize forest carbon sequestration services and facilitate in precisely estimating the forest carbon sink.

# INTRODUCTION

The value of urban forests in mitigating carbon emissions has received considerable interest among scientists and urban planners (*Zhao et al., 2010*; *Setala et al., 2013*) because of accelerated urbanization and ecological and environmental problems worldwide (*Maruotti, 2008*; *Mccarthy, Best & Betts, 2010*; *Zhao et al., 2015*; *Zhao & Wentz, 2016*). The carbon sink function of urban forests in many cities such as Miami-Dade and Gainesville, USA (*Escobedo et al., 2010*); Hangzhou, China (*Zhao et al., 2010*); and Chuncheon, Kangleung, and Seoul, Middle Korea (*Jo, 2002*) have been assessed for offsetting regional carbon emissions. Even though researchers worldwide have estimated the storage and sequestration of carbon in urban forests (*Liu & Li, 2012*; *Nowak et al., 2013*; *Zhang et al., 2015*), less attention has been paid to the relationship between the landscape pattern and carbon storage functions in urban forests (*Ren et al., 2013*). In general, urban regions have limited space for urban forests, and the landscape patterns of urban forests could strongly modify forest ecological services, such as species diversity conservation (*Zhang et al., 2017b*) and cooling island effects (*Ren et al., 2013*). To date, associations between carbon sinks and landscape patterns have not been well-defined, and untangling the associations is essential to precisely evaluate ecological services of urban forests (*Ren et al., 2013*; *Zhang et al., 2017b*; *Ren et al., 2018*; *Wang et al., 2018*).

Quantification of forest landscape metrics is useful for understanding the structure and configuration of the urban forest landscape (*Uuemaa, Mander & Marja, 2013*). Possible indices include area-related parameters, such as the total area for green infrastructure (TA), number of forest patches (NP), and mean patch size (AREA_MN); forest shape features, such as perimeters of patch and perimeter-area ratio (PARA); patch aggregation features, such as landscape shape index (LSI), largest patch index (LPI), and distance between patches (*Liu et al., 2009*; *Gounaridis, Zaimes & Koukoulas, 2014*). These indices yield a lot of information and can be quantified easily by using ArcGIS and Fragstats software; they have been used as indicators in many studies on land use changes, habitat alterations, and landscape regulating functions under both urban and natural conditions (*Uuemaa, Mander & Marja, 2013*; *Ren et al., 2013*; *Zhang et al., 2017b*). Associations between urbanization intensity (urban–rural gradients) and ecological services, such as carbon storage capacity, have been used to understand urbanization and green infrastructure
service interactions (*Zhang et al., 2015*; *Lv et al., 2016*) without considering forest landscape patterns. Clarification of the relationships between urban forest landscape characteristics and carbon storage may be helpful in understanding how urbanization affects carbon storage in trees and soils and finding possible indicators for evaluations of urban forest carbon sink functions; the possible parameters include urbanization intensity, landscape metrics, and tree size and forest community attributes, etc. (*Wang et al., 2005*; *Wang et al., 2011*; *Zhang et al., 2017b*).

Currently, China is in a period of rapid urbanization, and there is a general hope for good environmental security from urban green vegetation, such as forests (*Mu, Sun & Zhu, 2004*; *Wang, 2009*; *Liang et al., 2014*; *He et al., 2017*; *Zheng et al., 2017*; *Wang et al., 2018*). In China, the urban forests can easily be classified into different types on the basis of their location and functions, such as roadside forest distributed on both sides of roads; affiliation forest in different units of universities, schools, and institutes; landscape and relaxation forest in gardens and parks; and ecological public welfare forest for the prevention of soil erosion, floods, etc. (*He et al., 2004*). The division of different administrative districts in the same city facilitates possible management by public and private units (*Liu, Li & Guo, 2007*), including urban vegetation. Moreover, ring-road development and the history of urban settlements indicating urban–rural gradients are typical ways for understanding the urbanization effects (*Xiao et al., 2016a*). Studies that combine forest types, administrative districts, and urban–rural gradients, together with suitable landscape metrics, may favor the quantification of urban forest landscape features and their associations with forest ecological processes (*Liu et al., 2009*; *Ren et al., 2013*) and provide possible strategies for forest city design during the fast urbanization process in China (*Zhang et al., 2017b*).

Harbin, a typical provincial capital city in Northeastern China, has been checked for urbanization effects on urban tree species diversity and possible associations with bird species alternations (*Xiao et al., 2016a*), tree species configuration problems in urban afforestation (*Xiao et al., 2016b*), and best tree species for improving urban forest soils (*Lu et al., 2016*; *Wang et al., 2017*) and soil glomalin and mycorrhizal features (*Zhong et al., 2016*; *Cui & Mu, 2016*). However, whole carbon storage at city scale, especially soil organic carbon (SOC), and the relationships with landscape pattern characteristics are still unknown (*Ying, Li & Fan, 2009*; *Lv et al., 2016*).

In this paper, we used Harbin, the northernmost province capital city in China, as an example and hypothesized that urban forests have largely different landscape metrics on the basis of the forest type, administrative district, and urban–rural gradients, and these differences are closely associated with forest carbon sequestration in the biomass and soils. The following questions have been answered in this paper: (1) How large are the variations in the characteristics of urban forest landscapes in Harbin City? What are the differences in forest types, administrative districts, and urban–rural gradients? (2) How large are the differences in urban forest biomass and underground soils? Which landscape metrics are most significantly associated with these carbon sequestration differences? (3) Is there any management suggestions from the view of landscape configurations for promoting carbon storage in urban forests? What type of landscape-related suggestions for the exact evaluation
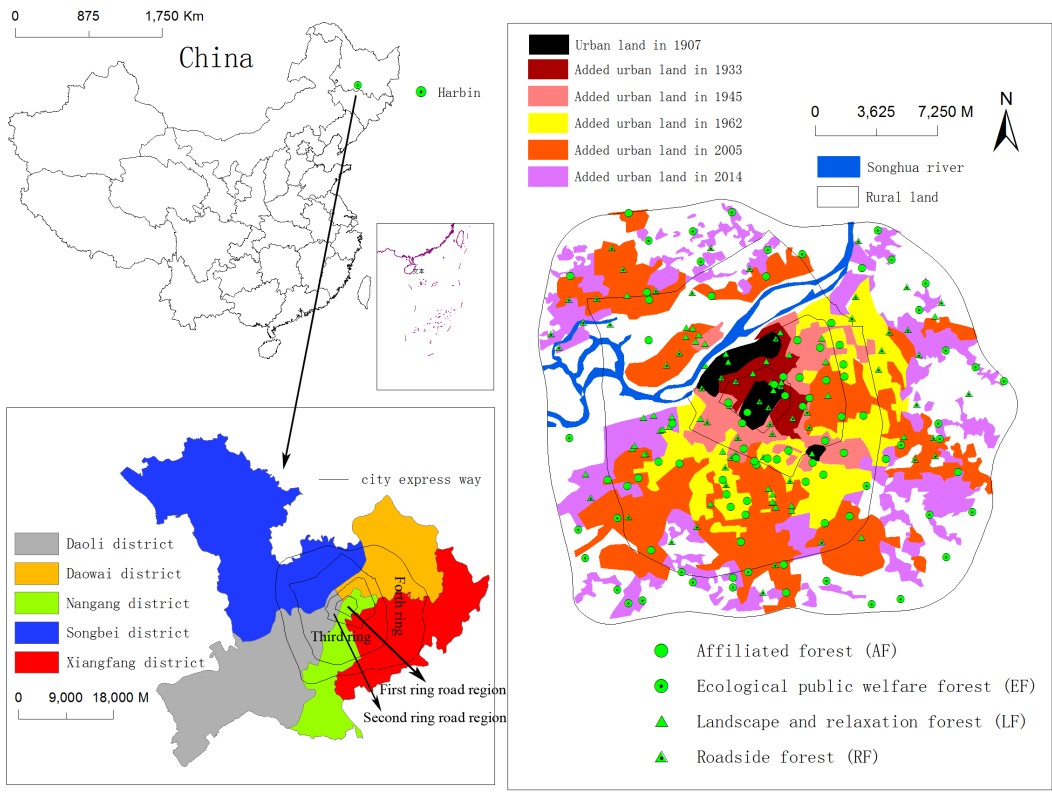

**Figure 1  Location of the study area showing Harbin City in northeastern China, and the distribution of sampling plots at different forest type, administrative districts and urban-rural gradients of different ring roads and history of urban settlements in Harbin.**

of urban forest carbon storage could be derived? The solutions will favor landscape-carbon sequestration sciences and help to define a suitable strategy for urban afforestation and management.

## MATERIALS & METHODS

### Study area

The study area is located in the urban area of Harbin City (45°45′N, 126°38′E; Fig. 1), the capital of Heilongjiang Province in Northeastern China. The average elevation of Harbin City is 151 m above sea level. The municipal district covers an area of 10,198 km². The total area within the fourth ring road is about 600 km², and the built-up area within the fourth ring road is 345.3 km². Until 2014, 4.7 million people lived in the downtown area, according to the Statistical Yearbook of Harbin (http://www.harbin.gov.cn/col/col39/index.html). The mean temperatures in January and July are −17.6 °C (0.3 °F) and 23.1 °C (73.6 °F), according to the climate data from 1981 to 2010 (http://data.cma.cn/site/index.html). The annual precipitation is 524 mm (20.6 in). The frost-free period lasts 140 days, while the

ice period lasts 190 days (*Zhang et al., 2011*). The most prevalent soil across Harbin is the black soil (Luvic Phaeozem, FAO) (*Chang, 2015*).

The main region in Harbin City is composed of five administrative districts. The main forest types are roadside forest (RF), ecological public welfare forest (EF), landscape and relaxation forest (LF), and affiliated forest (AF), according their location, ecological function, and management objectives (*He et al., 2004*). Long-term records (up to 1900s) on the urbanization process in Harbin are available. The urban land in Harbin was only 12 km$^2$ in 1907, and it had increased ca. 30-fold to 333 km$^2$ in 2014 (Fig. 1). The urban population in Harbin increased 18-fold from 0.54 million in 1946 to 9.87 million in 2014 (*Yi et al., 2006*). The expansion of ring roads is also a good substitute for urban–rural gradients in Harbin (*Huang et al., 2010*). These features made Harbin City a good example for the study of urbanization effects on urban forest landscape characteristics and their associations with carbon storage capacity.

## Image processing and tree coverage interpretation

Two multispectral GF-1 images with a resolution of 2 m × 2 m (China Center for Resources Satellite Data and Application; CRESDA) acquired on September 26, 2014, were used for the forest coverage extraction. Image preprocessing procedures included radiometric correction, FLAASH atmospheric correction, ortho-rectification, and image fusion and clipping. We extracted the urban forest coverage by using the object-oriented based classification method, according to texture information, spectral information, and spatial attributes of remote sensing images. The attribute assignment and manual modifications were performed with ArcGIS map (Esri, version 10.0; Redlands, CA, USA).

We used per pixel (*Wentz & Zhao, 2015*) and object-based (*Li et al., 2015*) validation methods to evaluate the accuracy of urban forest tree coverage extraction. Per pixel methods are used to check whether an individual pixel is classified correctly (*Wentz & Zhao, 2015*). The overall accuracy of the tree coverage extraction was 97.67%, with a miss factor of 0.06 and detection rate and quality percentage of 94.21%. Object-based accuracy assessment was performed using manually digitalized results from Google Earth satellite images (resolution, 0.59 m) as reference data (*Li et al., 2015*). We randomly selected 100 plots (500 m × 500 m) to compare the discrepancies between the classified results and reference maps. Figure 2 shows the scatter plot of tree coverage between classification results and corresponding reference data (Google Earth image). It is evident that these scattered points are distributed near the 45° line and $r^2$ is more than 0.99. The results for forest tree coverage in Harbin City are presented in Fig. 3.

## Analysis of landscape metrics

The analysis of forest landscape characteristics was based on four forest types (*He et al., 2004*), five administrative districts, four ring roads, and seven periods of urban settlements (*Chen et al., 2005*; *Zhang et al., 2015*) and performed using Fragstats (v4.2.589; Fig. 3). The landscape metrics were selected according to their ecological meanings and referenced from *Liu et al. (2009)* and *Gao & Yu (2014)*. Seven indices (Table SA1) were finally selected. The area and edge indices were total area (TA), number of patches (NP), largest patch index
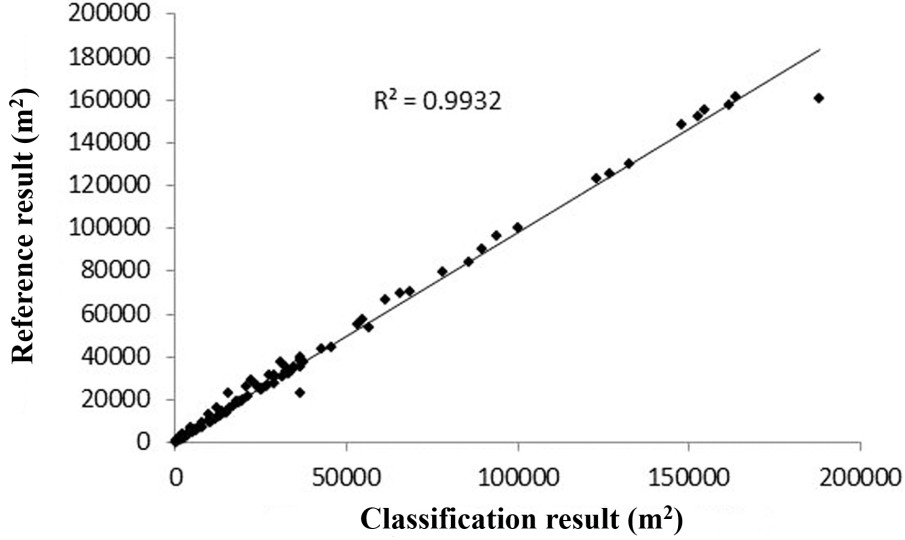

**Figure 2** Precision validation for tree cover data of urban forests in Harbin City using Google Earth images with a resolution of 0.59 m.

(LPI), and mean patch area (AREA_MN); shape index was area-weighted perimeter-area ratio (PARA_AM); and aggregation indices were landscape shape index (LSI) and mean Euclidean nearest-neighbor distance (ENN_MN).

The relationships between patch perimeter and patch area are the bases for most shape indices, and the Euclidean nearest-neighbor distance (ENN) is the simplest measure of patch context. We also analyzed the frequency distribution of four indices at patch level, namely, patch area, patch perimeter, perimeter-area ratio (PARA), and ENN. The spatial scale for analysis was 2 m, the same as the GF-1 image resolution, and the eight-neighbor patch rule was used.

## Field study and estimation of carbon storage

The field study was conducted between August and September 2014. During the field study, we recorded the location of each plot (by using GPS), tree species composition, diameter at breast height (DBH; 1.3 m), basal area (1.3 m, at breast height), and height (measured using a Laster tree height meter, Nikon forestry PRO550 Nikon, Japan) of trees with a diameter greater than 1 cm. The soil samples (0–20 cm) were collected at the same time by using a 100 cm$^3$ cutting ring (four cutting rings per plot; M&Y Instrument Technology Co. Ltd., Shanghai, China). A fixed volume of intact soil (400 cm$^3$) was stored in a cloth soil bag and air-dried in a dry ventilated room to constant weight for laboratory analysis.

The sampling plots were allocated according to forest coverage in different forest types, administrative districts, ring roads, and history of urban settlements (Fig. 1). Plot allocation is listed in Table 1. The estimation of total carbon storage was the random sampling statistics of carbon storage density (carbon stocks per tree cover) of different forest types, ring roads, administrative districts, and history of urban settlements (*Lv et al., 2016*) and their tree

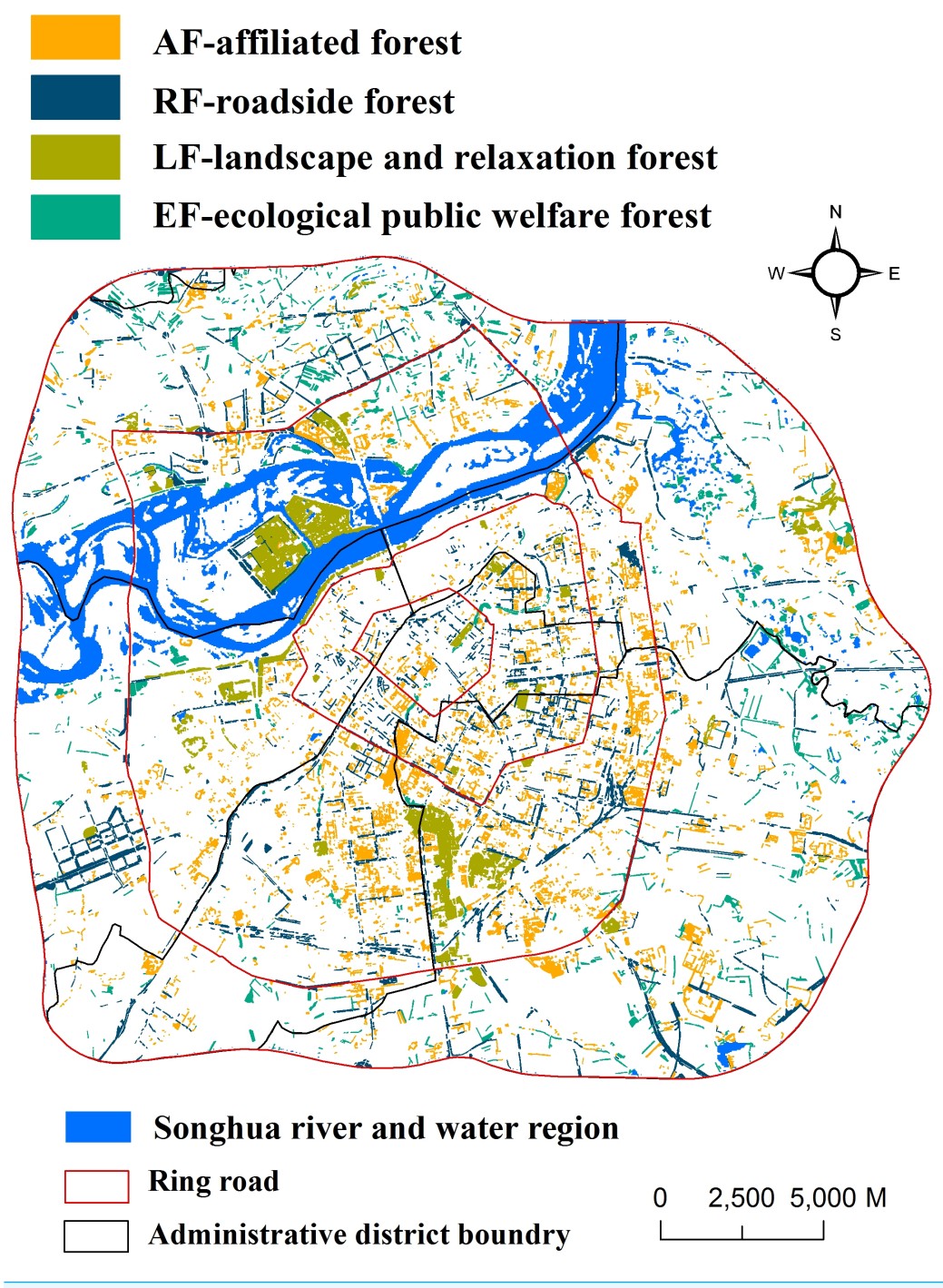

AF-affiliated forest

RF-roadside forest

LF-landscape and relaxation forest

EF-ecological public welfare forest

Songhua river and water region

Ring road

Administrative district boundry

0    2,500  5,000 M

**Figure 3** Spatial distribution of urban forests in Harbin City derived from GF1 images (2 m resolution).

**Table 1  Trees and soils carbon storage for different forest types, administrative districts, urban-rural gradients (ring roads and history of urban settlements) in Harbin City.**

| Urban forests classification | | No. of plots | Area of the region km² | Carbon storage (thousand tons) | | | C storage density (tons ha⁻¹) | |
|---|---|---|---|---|---|---|---|---|
| | | | | Tree | Soil | Total | Tree biomass | Soil |
| Different forest types and regions | | | | | | | | |
| Forest types | AF | 58 | 12[a] | 68.9 | 63.7 | 132.6 | 51.4 (6.4) | 47.5 (3.4) |
| | RF | 42 | 9[a] | 72.6 | 56.2 | 128.8 | 71.6 (10.6) | 55.4 (3.1) |
| | LF | 62 | 10[a] | 60.9 | 66.3 | 127.2 | 62.9 (8.6) | 68.5 (5.5) |
| | EF | 36 | 6[a] | 100.0 | 32.7 | 132.7 | 155.0 (12.2) | 50.7 (3.6) |
| Administrative districts | Daoli | 34 | 90 | 18.5 | 26.1 | 44.6 | 46.2 (8.3) | 65.2 (6.8) |
| | Daowai | 30 | 96 | 52.7 | 29.8 | 82.5 | 100.0 (1.9) | 56.5 (4.3) |
| | Nangang | 49 | 92 | 51.5 | 38.0 | 89.5 | 77.5 (8.3) | 57.2 (3.9) |
| | Songbei | 34 | 143 | 64.8 | 43.3 | 108.1 | 66.1 (8.4) | 44.1 (4.1) |
| | Xiangfang | 50 | 167 | 144.8 | 74.2 | 219.0 | 104.0 (13.3) | 53.3 (3.4) |
| Urban-rural gradients | | | | | | | | |
| Ring roads | First ring | 16 | 11 | 4.8 | 4.8 | 9.6 | 68.8 (13.8) | 69.1 (4.9) |
| | Second ring | 32 | 48 | 22.7 | 21.0 | 43.7 | 60.8 (8.1) | 56.1 (5.7) |
| | Third ring | 77 | 205 | 105.1 | 102.8 | 207.9 | 56.1 (5.8) | 54.9 (3.4) |
| | Fourth ring | 74 | 328 | 178.1 | 83.4 | 261.5 | 108.1 (1.2) | 50.6 (3.0) |
| History of urban settlements | 100-yr | 7 | 12 | 3.8 | 5.2 | 9.0 | 50.3 (16.5) | 69.4 (6.3) |
| | 80-yr | 10 | 13 | 5.4 | 4.2 | 9.6 | 78.7 (15.8) | 60.6 (9.0) |
| | 70-yr | 28 | 32 | 19.7 | 19.5 | 39.2 | 62.6 (9.2) | 62.0 (5.1) |
| | 50-yr | 27 | 62 | 34.8 | 25.7 | 60.5 | 68.4 (14.9) | 50.6 (5.2) |
| | 10-yr | 44 | 138 | 82.4 | 75.8 | 158.2 | 59.6 (9.8) | 54.8 (3.3) |
| | 0-yr | 51 | 92 | 25.0 | 23.0 | 48.0 | 55.2 (8.2) | 50.7 (5.0) |
| | unsettled | 32 | 242 | 188.0 | 62.2 | 250.2 | 161.8 (13.4) | 53.5 (3.9) |
| Sum | | 199 | | 302–359 | 211–219 | 521–575 | | |

**Notes.**

AF, affiliated forest; RF, roadside forest; LF, landscape and relaxation forest; EF, ecological public welfare forest; 100-yr, urban area constructed before 1906; 80-yr, urban area constructed between 1933 and 1907; 70-yr, urban area constructed between 1945 and 1934; 50-yr, urban area constructed between 1962 and 1946; 10-yr, urban area constructed between 2005 and 1963; 0-yr, urban area constructed during 2006 and 2014; unsettlement, rural land.

The numbers in brackets are standard errors.

[a] Tree coverage area.

coverage area shown in Fig. 3.

$$\text{Carbon}_{storage} = \sum_{i=1}^{j} CD_i \times TC_i \qquad (1)$$

where $CD_i$ is the $i$th carbon storage density of different forest types, ring roads, administrative districts, and history of urban settlements, and $TC_i$ is the $i$th urban forest tree coverage area of these urban forest classifications.

The dryweight biomass of the trees was estimated using tree biomass allometric growth equations obtained from published literature and root-to-shoot ratio of 0.26 when belowground biomass equations were absent (Table SA2). Total tree dryweight biomass was converted to total stored carbon by multiplying by 0.5. The biomass carbon storage density

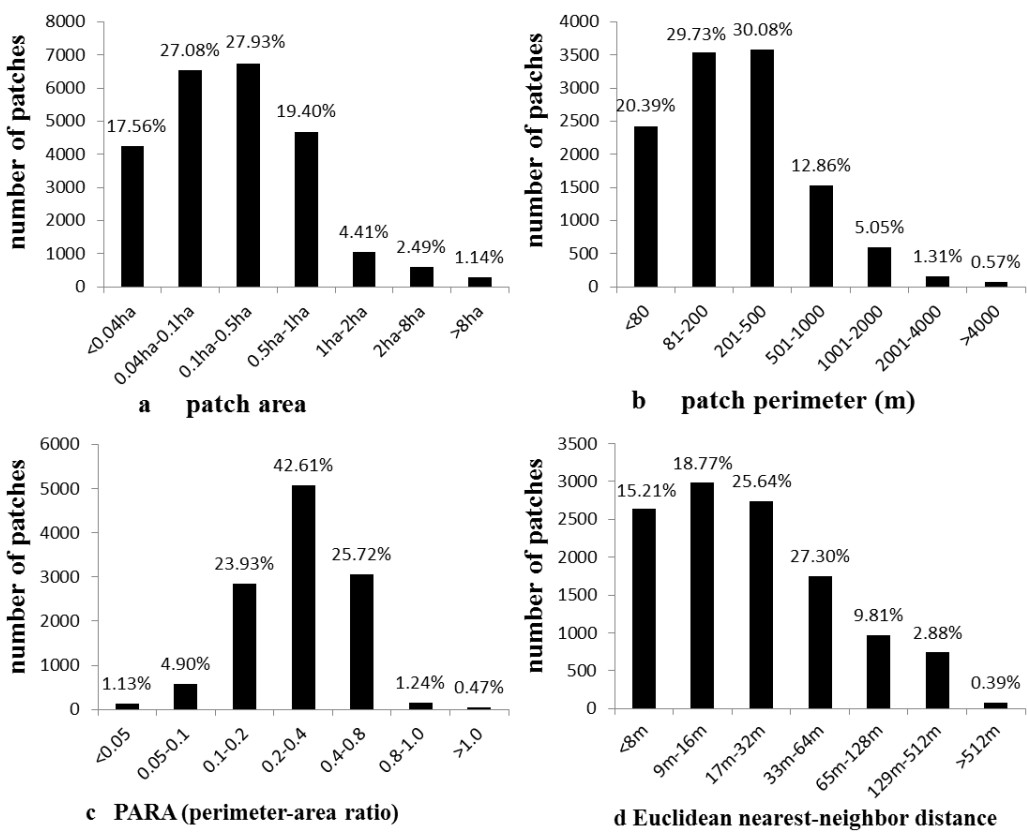

**Figure 4** Frequency distribution of urban forests patch area (A), perimeter (B), perimeter-area ratio (C), and Euclidean nearest neighbor distance (D) in Harbin City.

(kg C × m$^{-2}$) was estimated using total tree biomass carbon in each plot, divided by the plot area.

The SOC content was determined using the heated dichromate/titration method (*Bao, 2000*; *Wang et al., 2011*). SOC density in each plot was the product of SOC content, soil bulk density, and sampling soil depth (20 cm in this study).

## Statistical analysis

The landscape patterns of the urban forests were characterized by the frequency distribution of different metrics of pooled whole-city data. For finding the changes at urban–rural gradients in ring road development and history of urban build-up, linear regression was used for detecting steady changes during urbanization processes.

Pearson's correlation, linear and stepwise regression analyses, smoothing analysis, and bivariate normal distribution were performed using SPSS (version 19.0, 2010, IBM, USA) and JMP (SAS, version 10; Chicago, IL, USA), respectively, to understand the associations between landscape characteristics and carbon stocks. All tables and Figs. 2, 4 and 5 were created with MS Excel 2010 (14.0.4760.1000; Microsoft, Redmond, WA, USA); Figs. 1 and 3 depicted using ArcGIS 10.0; Fig. 6 depicted using JMP 10.0.

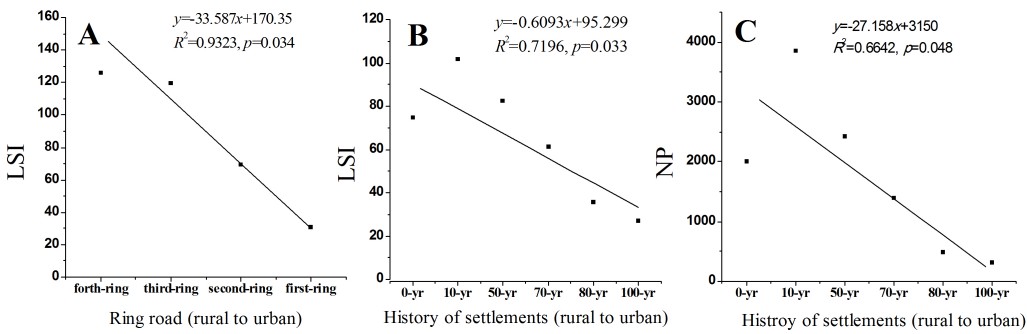

**Figure 5** Changes of landscape shape index (LSI) and number of patches (NP) of urban forests along urban-rural gradients (ring roads or history of settlements) in Harbin City.

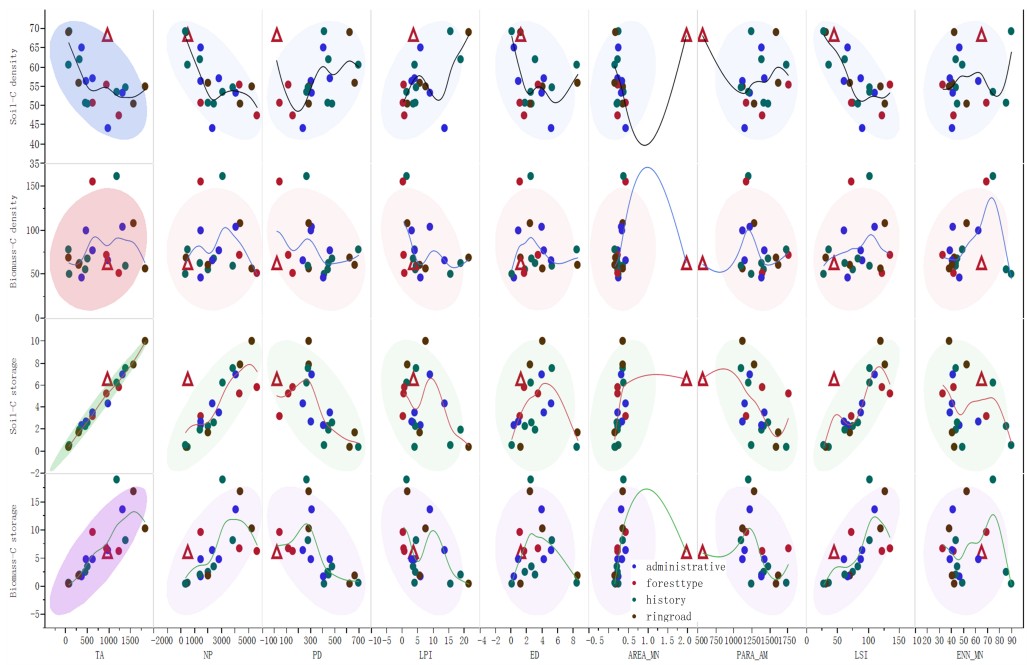

**Figure 6** Associations between different landscape metrics and soil-C density, biomass-C density, soil-C storage as well as biomass-C storage. Note: The line in the figure is the smoothed line of the raw data. The ellipse in the figure is the bivariate normal distribution of the raw data. The triangle is the LF data, which is much larger than others in Area-MN, and without LF, area-MN had good linear relations with various carbon parameters. The different colors of the labels in the figure showed that the data originated from administrative districts, forest types, urban history and ringroad development regions.

# RESULTS

## Spatial distribution and patch characteristics of urban forests

The spatial distribution of urban forests in Harbin was highly uneven (Fig. 3). The large patches mainly belonged to LF, and RF was mainly a long rectangular strip. AF was mainly aggregated in the central urban regions, whereas EF was mainly located in the outer rural

region of Harbin City (Figs. 1 and 3). With respect to urban–rural gradients, large patches were mainly distributed in the third ring road region, forest patches in the second ring road region were mainly aggregated in the south area, and north regions of the second ring road had little forest cover. The urban forest in the oldest central urban regions was mainly AF. The urban forests in the newest urbanized regions, such as the north side of Songhua River, were mainly EF. In different administrative districts, most patches in Songbei District were aggregated along Songhua River, and most patches in Xiangfang and Daowai Districts were aggregated in the forest botanical garden and Tianhengshan Park. In summary, the forest patches in all administrative districts were aggregated in several separate areas and most areas were treeless.

The forest patches in Harbin were highly fragmented. Small patches covering an area of less than 0.5 ha and patches covering an area of 0.5 to 1 ha accounted for 72.6% and 19.4% of all patches, respectively. Large patches covering an area of more than 1 ha accounted for only 8% of all patches (Fig. 4A). The average patch size was 0.31 ha. Patches with a perimeter of <500 m accounted for 80.2% and patches with a perimeter of more than 1,000 m accounted for only 6.9% of all patches (Fig. 4B). The average perimeter of the patches was 388 m. Patches with PARA of more than 0.8 and less than 0.01 accounted for only 7.7% of all patches. More than 90% of the patches had PARA of 0.1 to 0.8, of which 42.6% had PARA from 0.2 to 0.4 (Fig. 4C); the average PARA value was 3251. Most patches (59.62%) were less than 32 m away from the nearest neighboring patches, and only less than 1% of the patches were isolated and more than 512 m away from other patches. ENN of the remaining patches ranged from 32 m to 512 m and they accounted for 39.99% of all patches (Fig. 4D); the average ENN value was 43 m.

## Urban forest carbon: forest types, administrative districts, and urban–rural gradients

The total carbon storage in the different forest types was almost equal, ranging from 127 to 133 thousand tons (Table 1). The tree carbon storage density was the highest in EF, and soil carbon storage density was the highest in LF.

The peak value for total carbon storage was detected in Xiangfang District (219 thousand tons), which was 4.9-fold higher than the lowest value in Daoli District. The peak value for tree biomass carbon density was observed in Xiangfang District (104 tons ha$^{-1}$), whereas the peak value for soil carbon density was detected in Daoli District (65 tons ha$^{-1}$; Table 1).

With respect to urban–rural changes, a general pattern of linear decrease from the urban center to rural regions was found with respect to soil carbon density, whereas no steady changes in tree biomass carbon density were found (Table 1). Carbon storage in the first and second ring roads accounted only for less than 10% of the total carbon storage in Harbin, and the highest carbon storage was detected in the fourth ring road (262 thousand tons). With respect to different history of urban settlements, the peak value for total carbon storage was in the unsettled region (250 thousand tons), which was 28-fold higher than the lowest value in the 100-year history region (nine thousand tons). The peak value for tree carbon density was also in the unsettled region, whereas soil carbon density increased along with the history of urban settlements. The peak value for soil carbon density was

**Table 2 Landscape metrics of different types, administrative districts, and urban-rural gradients (ring roads and history of settlements) of urban forests in Harbin City.**

| Urban forests classification | | TA (ha) | NP | LPI (%) | AREA_MN (ha) | PARA_AM | LSI | ENN_MN (m) |
|---|---|---|---|---|---|---|---|---|
| Different forest types and regions | | | | | | | | |
| Forest types | AF | 1,226 | 5,631 | 0.69 | 0.22 | 1,394 | 121.99 | 41.63 |
| | RF | 942 | 4,327 | 0.41 | 0.22 | 1,752 | 134.38 | 32.42 |
| | LF | 959 | 488 | 3.59 | 1.97 | 559 | 43.28 | 64.57 |
| | EF | 620 | 1,463 | 0.28 | 0.42 | 1,165 | 72.47 | 68.89 |
| Administrative districts | Daoli | 364 | 1,477 | 5.78 | 0.25 | 1,385 | 65.98 | 46.70 |
| | Daowai | 474 | 1,435 | 2.96 | 0.33 | 1,116 | 60.70 | 62.54 |
| | Nangang | 614 | 2,803 | 3.99 | 0.22 | 1,420 | 87.67 | 38.57 |
| | Songbei | 981 | 2,315 | 13.75 | 0.42 | 1,151 | 89.74 | 40.59 |
| | Xiangfang | 1,309 | 4,038 | 8.89 | 0.32 | 1,216 | 109.62 | 41.15 |
| Urban-rural gradients | | | | | | | | |
| Ring road regions | First ring | 60 | 372 | 21.55 | 0.16 | 1,582 | 30.55 | 42.12 |
| | Second ring | 299 | 2,001 | 5.65 | 0.15 | 1,620 | 69.65 | 40.19 |
| | Third ring | 1,827 | 5,224 | 7.39 | 0.35 | 1,121 | 119.38 | 38.20 |
| | Fourth ring | 1,561 | 4,392 | 1.65 | 0.36 | 1,278 | 125.93 | 52.78 |
| History of urban settlements | 100-yr | 75 | 311 | 15.44 | 0.24 | 1,242 | 26.93 | 89.64 |
| | 80-yr | 69 | 482 | 3.99 | 0.14 | 1,725 | 35.67 | 49.00 |
| | 70-yr | 315 | 1,404 | 18.86 | 0.22 | 1,385 | 61.27 | 43.60 |
| | 50-yr | 508 | 2,418 | 3.47 | 0.21 | 1,468 | 82.57 | 44.93 |
| | 10-yr | 1,382 | 3,858 | 4.36 | 0.36 | 1,100 | 101.71 | 43.31 |
| | 0-yr | 453 | 2,008 | 4.12 | 0.23 | 1,407 | 74.77 | 85.30 |
| | unsettled | 1,169 | 3,067 | 1.44 | 0.38 | 1,196 | 102.02 | 74.16 |

**Notes.**

AF, affiliated forest; RF, roadside forest; LF, landscape and relaxation forest; EF, ecological public welfare forest; 100-yr, urban area constructed before 1906; 80-yr, urban area constructed between 1933 and 1907; 70-yr, urban area constructed between 1945 and 1934; 50-yr, urban area constructed between 1962 and 1946; 10-yr, urban area constructed between 2005 and 1963; 0-yr, urban area constructed during 2006 and 2014; unsettlement, rural land; TA, total area; NP, number of patches; LPI, largest patch index; AREA_MN, mean patch area; PARA_AM, Area mean Perimeter-Area Ratio Distribution; LSI, Landscape Shape Index; ENN_MN, Mean Euclidean Nearest Neighbor Distance Distribution.

in the 100-year region (soil carbon density $= 0.15 \times$ (year of settlement history) $+ 50$, $R^2 = 0.70$, $p < 0.05$, according to our previous study (*Lv et al., 2016*).

## Landscape metrics: forest types, administrative districts, and urban–rural gradients

AF comprised the largest proportion of all forest types, with a TA of about 2 times that of EF. The TAs of LF and RF were almost equal. LF had a low level of fragmentation, with the largest LPI (3.59%). The patch size (AREA_MN, 1.97 ha) in LF were 5–12-fold and 5–9-fold higher than that in the other forest types, and the lowest PARA_AM (559) and LSI (43.28) in LF were only half of those in the other forest types. EF had the lowest LPI (0.28%) and the highest ENN_MN (68.89 m); AF and RF had a high level of fragmentation, and they had the lowest AREA_MN (0.22 ha) and higher PARA_AM and LSI than the other forest types (Table 2).

The highest TA of urban forests (1,309 ha) and NP (4,038) in Xiangfang District were 3.6-fold higher than the lowest TA in Daoli District and 2.8-fold higher than the lowest NP in Daowai District. The largest LPI (13.75%) and AREA_MN (0.42 ha) in Songbei District were 4.6-fold higher than the lowest LPI in Daowai District and 1.9-fold higher than the lowest AREA_MN in Nangang District. The peak value of ENN_MN (62.54 m) in Daowai District was 1.6-fold higher than that of the lowest value in Nangang District (38.57 m; Table 2).

The TA of the suburb urban regions (third and fourth ring roads; 3,388 ha) was 9-fold higher than that of the central urban regions (first and second ring roads; 359 ha). The mean patch size (<0.16 ha) in the central urban area was half of that in the suburb area, and PARA_AM in the central urban area was higher than that in the suburban regions (Table 2). The dispersion of patches (LSI) linearly decreased from the fourth ring road region to the first ring road region (LSI $= -33.587 \times$ ring road $+ 170.35$, $R^2 = 0.93$, $p = 0.034$; Fig. 5A).

The NP and LSI linearly decreased from the 0-year history region (rural region) to the 100-year region (urban center) (LSI $= -0.6093 \times$ history $+ 95.299$, $R^2 = 0.7196$, $p = 0.033$; NP $= -27.158 \times$ history $+ 3,150$, $R^2 = 0.6642$, $p = 0.048$; Figs. 5B and 5C). The younger history regions like the 10-year region had the highest TA (1,382 ha), NP (3,858), and AREA_MN (0.36 ha), and they were 20-fold and 2.6-fold higher than the lowest TA (69 ha) and AREA_MN (0.14 ha) in the 80-year region and 12-fold higher than the lowest NP (311) in the 100-year region (Table 2).

**Associations between forest carbon traits and landscape metrics**

As shown in Table 3 and Fig. 6, Pearson's correlation analysis, smoothing analysis, and bivariate normal distribution were used to show the associations between the carbon parameters and landscape metrics. The total tree and soil carbon storage values were positively correlated with TA, NP, AREA_MN, and LSI ($p < 0.01$). Of these 4 indices, peak coefficients were found in TA with tree biomass carbon (0.986) and soil carbon (0.806). The biomass and soil carbon density showed different associations with landscape metrics. When the LF data (outside the bivariate normal distribution ellipse) was excluded, the biomass carbon density was positively correlated with AREA_MN ($p < 0.05$, Table 3, Fig. 6), whereas no marked associations were found with the other indices ($p > 0.05$). The soil carbon density was negatively correlated with TA, NP, AREA_MN, and LSI (Table 3, Fig. 6) and positively correlated with LPI ($p < 0.05$, Table 3, Fig. 6). The highest association was found with LPI. No marked relationship was found between soil carbon density and PARA_AM ($p > 0.05$, Table 3, Fig. 6).

The stepwise regression analysis showed that both tree carbon storage and soil carbon storage were mainly associated with TA, whereas other factors such as ENN-MN and AREA-MN were included in the regression model (Table 4). When the standard coefficients were compared, the contribution of TA to carbon storage was 3.5–8.1-fold higher than that of EN-MN or AREA-MN. The stepwise regression analysis showed that the soil carbon storage density was mainly associated with LSI (soil carbon storage density $= -0.158 \times$ LSI $+ 69.316$, $r^2 = 0.51$, Table 4), whereas tree carbon storage density was associated with LPI
**Table 3 Pearson correlation analysis between landscape metrics and biomass and soil carbon storage (thousand tons) of urban forests in Harbin City.** Number of sample size = 20.

| Items | Pearson correlation | TA (ha) | NP | LPI (%) | AREA_MN (ha) | PARA_AM | LSI | ENN_MN (m) |
|---|---|---|---|---|---|---|---|---|
| Storage (ton) | Tree C | 0.986[**] | 0.782[**] | −0.388 | 0.608[**] | −0.505[*] | 0.769[**] | −0.238 |
| | Soil C | 0.806[**] | 0.609[**] | −0.422 | 0.716[**] | −0.370 | 0.687[**] | 0.026 |
| Density (ton ha$^{-1}$) | Tree C | 0.196 | 0.043 | −0.353 | 0.533[*] | −0.154 | 0.180 | 0.276 |
| | Soil C | −0.518[*] | −0.644[**] | 0.466[*] | −0.503[*] | −0.060 | −0.713[**] | 0.182 |

Notes.
[*] 0.05 significant level.
[**] 0.01 significant level.
 TA, total area; NP, number of patches; LPI, largest patch index; AREA_MN, mean patch area; PARA_AM, Area mean Perimeter-Area Ratio Distribution; LSI, Landscape Shape Index; ENN_MN, Mean Euclidean Nearest Neighbor Distance Distribution.

**Table 4 Stepwise regression between forest carbon parameters and landscape metrics.** Stepwise Criteria: Probability-of-F-to-enter ≤ 0.200, Probability-of-F-to-remove ≥ 0.300.

| Response variables | Parameters | Unstandardized coeff. | | Standardized coeff. | $t$-value | Sig. | $R^2$ |
|---|---|---|---|---|---|---|---|
| | | B | Std. error | Beta | | | |
| Density | | | | | | | |
| SOC density | (Constant) | 69.316 | 3.183 | | 21.780 | 0.000 | 0.508 |
| | LSI | −0.158 | 0.037 | −0.713 | −4.312 | 0.000 | |
| Biomass-C density | (Constant) | 90.092 | 10.161 | | 8.866 | 0.000 | 0.125 |
| | LPI | −1.848 | 1.153 | −0.353 | −1.603 | 0.126 | |
| Storage | | | | | | | |
| Biomass-C storage | (Constant) | −4.702 | 2.847 | | −1.651 | 0.117 | 0.708 |
| | TA | 0.009 | 0.001 | 0.870 | 6.414 | 0.000 | |
| | ENN_MN | 0.081 | 0.044 | 0.249 | 1.838 | 0.084 | |
| SOC storage | (Constant) | −0.051 | 0.150 | | −0.342 | 0.736 | 0.986 |
| | TA | 0.005 | 0.000 | 0.959 | 32.050 | 0.000 | |
| | AREA_MN | 0.849 | 0.215 | 0.118 | 3.956 | 0.001 | |

(tree carbon storage density $= -1.848 \times$ LPI $+ 90.902$, $r^2 = 0.125$, Table 4). These data showed that the carbon density was regulated to a greater extent by patch configuration and aggregation/dispersion than storage (Table 4).

## DISCUSSIONS

By 2050, there will be six billion urban dwellers (*Mccarthy, Best & Betts, 2010*). Urbanization, together with climate change, has become the biggest environmental problem worldwide, resulting in increased carbon emissions (*Maruotti, 2008*; *Mccarthy, Best & Betts, 2010*) and urban heat island effects (*Zhao et al., 2015*; *Zhao & Wentz, 2016*). Urbanization also affects landscape fragmentation and configuration and diversity of forests (*Su et al., 2012*) and significantly influences the structure, process, and ecological functions of urban vegetation ecosystems (*Yin et al., 2009*), including carbon storage function (*Zhang et al., 2015*; *Lv et al., 2016*). Urban forest landscape changes along urban–rural gradients are not well-defined, although many ecological functions, such as heat island, biodiversity, tree

size, and community features, have been discussed (*Larondelle & Haase, 2013*; *Lv et al., 2016*; *Xiao et al., 2016a*; *Zhang et al., 2017b*). Our study has supplemented information on forest landscape metrics and carbon storage in different forest types, administrative districts, and urban–rural gradients (in terms of ring road and settlement history) for better understanding of landscape characteristics under urbanization effects to provide suitable management measures targeted at different regions. The importance of our findings will be discussed below, together with reference comparisons, management suggestions, and future urban forest evaluations.

## Landscape fragmentation of urban forests in Harbin City: quantification and comparison

On basis of the overall trend for landscape fragmentation worldwide, fragmentation of the urban forest has been observed to be severe (*Liu & Zhang, 2012*; *Gong et al., 2013*). We used Harbin City as an example and quantified and compared landscape characteristics of urban forests in cities localized and worldwide. The mean patch size at city scale was 0.31 ha in Harbin, which was 29.8% of the forest patch size in Changchun (*Zhang, 2015*) and was low when compared with Shenyang (0.22–1.04 ha) (*Liu et al., 2009*). However, patch density (PD) and mean perimeter-area ratio (PARA) in Harbin, which reached up to 3.18 patches ha$^{-1}$ and 3,251, respectively, were both more than three-fold higher than those in Changchun (*Zhang, 2015*) and high when compared with Shenyang (PD and PARA ranged from 0.99 to 2.87 patches ha$^{-1}$ and 887 to 4,109, respectively) (*Liu et al., 2009*). These landscape characteristics all indicated a much higher fragmentation level of the urban forests in Harbin than in the localized cities in Northeastern China.

Worldwide urbanization has resulted in broad forest fragmentation. In Puerto Rico, forests became more fragmented between 1991 and 2000; the mean patch size decreased and edge to area ratio increased, the dynamics of forest fragmentation were synchronized with the urban sprawl, and the peak forest fragmentation shifted towards the rural areas (*Gao & Yu, 2014*). In Atlanta, Georgia, forests displayed a fragmentation trend from 1974 to 2005, and this fragmentation trend adversely affected habitat integrity (*Miller, 2012*). The fragmentation trends caused by urbanization have formed the current spatial structure of urban forests in Atlanta, Georgia (*Miller, 2012*), as in the urban forests in Harbin. The dispersion of patches in terms of LSI and number of patches (NP) decreased along the rural–urban gradients (Fig. 5), showing that forests in the rural regions of Harbin were more dispersed and had more patches than those in the central urban regions. Spatial–temporal gradient analysis of urban green spaces in Jinan also showed that the LSI of residential green space decreased along the rural–urban gradients, and together with other indices, affected urbanization (*Kong & Nakagoshi, 2006*).

Urban forest fragmentation was induced and influenced by many factors, such as urban building density (*Liu & Zhang, 2012*); deforestation and reforestation processes during urban sprawl (*Gao & Yu, 2014*); socioeconomic factors such as urban structure change, industry-related economic boom, increase in migrant resident population; and increased income of city residents (*Gong et al., 2013*); economically driven intense anthropogenic activities; and the absence of a sustainable environmental management and

conservation strategy (*Gounaridis, Zaimes & Koukoulas, 2014*). The fragmented landscape may degrade habitat quality; threaten species richness, abundance, and composition (*Iida & Nakashizuka, 1995*; *Liu et al., 2005*); and affect phylogenetic diversity (*Matos et al., 2016*). This may be a huge risk to forest management in cities such as Harbin. Harbin has experienced an exponential economic growth since the 1980s and population boom since the 1950s (*Xiao et al., 2016a*), resulting in insufficient greenery services in urban regions. The largely fragmented forest landscapes and uneven distribution in different regions of Harbin City require new efforts for urban forest protection, new plantation afforestation, and sustainable management activities for maximizing urban forest services.

## Landscape metrics responsible for carbon variations and high carbon-oriented managements

Urban forests can provide various ecosystem services and values to a city and its residents, like the removal of air pollutants (*Livesley, McPherson & Calfapietra, 2016*), alleviation of the urban heat island effect (*Zhao et al., 2018*), and reduction and offset of carbon emissions (*Jim & Chen, 2009*; *Nowak et al., 2013*). These ecological functions are linked to landscape characteristics based on the common consensus that environmental patterns strongly influence ecological processes (*Turner, 1989*; *Uuemaa, Mander & Marja, 2013*). Our findings highlighted that carbon storage function and landscape pattern of urban forests in Harbin are linked and landscape regulation is possible to improve urban vegetation and soil carbon storage. The carbon storage function of urban forests could contribute to the alleviation of climate changes (*Nowak et al., 2002*) and reduction of the negative effects of fast urbanization (*De Jong et al., 2015*). The increase of carbon storage at individual tree and patch scale is possible through tree health promotion, forest structure adjustment, tree species selection (*Nowak et al., 2002*), and soil improvement (*Jandl et al., 2007*). With respect to landscape pattern, it is important to increase carbon storage through landscape regulation, which should be an aspect of promoting forest ecological function in limited urban areas (*Ren et al., 2013*; *Lv et al., 2016*).

First, carbon storage in the urban forest trees (and soils) was positively correlated with NP and AREA_MN (Table 3, Fig. 6), which was consistent with the results reported by *Wang (2012)*. In a given urban green space, relatively large forest patches with considerable patch numbers are possible for increasing the urban forest carbon storage in the trees and soils. The much closer relationship between tree carbon storage and TA (Table 3, Fig. 6) indicated that an increase in forest cover is the best way to promote carbon storage function, especially in regions with rather low forest coverage. The increase in total forest coverage could also lower the degree of landscape fragmentation (*Liu & Zhang, 2012*) and enhance the cool island effect of urban forests (*Ren et al., 2015*).

Second, because of limited green spaces in cities, improvement of carbon storage density (tons ha$^{-1}$) through landscape regulations should be more practical and focus on forest construction, regulation, and functional promotion. For example, we found that the mean patch size and tree carbon storage density were positively correlated and increasing the mean patch size may promote tree carbon density in future landscape designs.

Third, LSI and LPI are two landscape metrics for carbon-oriented landscape regulation. Our data have shown that LSI and LPI are closely associated with carbon storage at city and plot density levels (Table 3, Fig. 6). LPI is the percentage of the largest patch in the landscape, mainly reflecting configuration. LSI mainly reflects the aggregation/dispersion of patches. Increasing the connectivity and aggregation of patches (decreasing LSI) and improving the promotion of the largest patch in landscape design may favor urban carbon sequestration.

Another parameter for facilitating carbon storage is AREA_MN, which was positively correlated with tree biomass carbon storage after the LF data were excluded (Table 3, Fig. 6). In future urban forest management, afforestation activities targeted at different regions are necessary, especially in regions with low AREA_MN such as Nangang District. Even though tree coverage in Harbin is low, together with other green spaces, green coverage in Harbin could reach up to 36% of the whole urban landscape (*Lv, 2017*). Afforestation in existing grassland, wasteland, and illegal construction land that unites small patches into larger and regular-shaped patches would increase the mean patch size and ecological services.

China has devoted a large financial budget for the construction of urban forests, and, by 2020, at least 200 cities will build close-to-nature forests based on the unified design between cities and the countryside to achieve the title of "National Forest City" (*Forestry, 2016*). On the basis of our findings, Harbin City still has a long way to go to be one of the 200 cities. The urban forest trees in Harbin cover only 7% of the urban lands, which is much lower than the lower limits for forest city (build-up region green coverage, 40% at least). In addition, the uneven distribution of urban forests in the different administrative districts and urban–rural gradients may hinder the unified design concept and lead to discrepancies in the ecological functions provided by urban vegetation in different regions. An increase in tree coverage at city scale is necessary to promote the overall ecological benefits of urban forests in Harbin, especially the establishment of a series of country, city, and community parks. These parks could benefit the local people as long as they open a window or door. Specific designs, such as demolition of illegal structures for tree planting, replanting trees in vacant lots that surround by buildings (*Bajsanski, Stojakovic & Jovanovic, 2016*; *Zhao, 2017*; *Zhao, Wentz & Murray, 2017*), and implementation of vertical greening on roofs, walls, and bridges, may increase the connectivity of urban forests and have a positive influence on the fragmented landscape (*Gao & Yu, 2014*). Previous studies have highlighted that urban greening practices should consider the importance of biodiversity conservation (*Xiao et al., 2016a*), removal of pollutants (*Escobedo & Nowak, 2009*; *Mu, Sun & Zhu, 2004*), urban microclimate regulations (*Wang et al., 2018*), urban heat island mitigation (*Zhang et al., 2017a*), and urban soil improvement (*Cui & Mu, 2016*; *Wang et al., 2017*; *Zhou et al., 2017*). Our findings strongly suggest that urban forest carbon sequestration in both aboveground biomass and belowground soil carbon should be considered for co-improvement of multiple ecological services.

## Implications for the urban forest evaluation and uncertainty

The total carbon storage of the urban forest within the forth ring road of Harbin City was 521 to 575 thousand tons, of which 302 to 359 thousand tons belongs to tree carbon;

this is more than the above 20 thousand tons estimated by *Ying, Li & Fan (2009)*. These findings show the large uncertainty in the estimation of urban forest carbon sequestration (*Pouyat, Yesilonis & Nowak, 2006*). Previous studies on the estimation of carbon storage in urban forests always focused on only vegetation biomass carbon and paid less attention to soil carbon (*Nowak et al., 2013*; *Zhang et al., 2015*). Urban soils have robust carbon storage capacity, both in the areas covered by green vegetation (*Liu et al., 2016*) and beneath the impervious surface (*Pouyat, Yesilonis & Nowak, 2006*; *Edmondson et al., 2012*; *Raciti, Hutyra & Finzi, 2012*). In the future, belowground soil carbon storage should be analyzed, and both aboveground and belowground carbon inclusion may help in understanding carbon storage capacity of the whole ecosystem.

The scaling up of tree biomass carbon storage in cities was based on several methods without considering landscape metrics, such as model-based estimation including UFORE, CITY green, i-TREE, and InVEST (*Nowak et al., 2013*); remote sensing image-based estimation by using carbon storage per pixel and vegetation index, such as NDVI (*Myeong, Nowak & Duggin, 2006*); forest inventory-based estimation (*Zhang, 2015*); and random sampling statistics for plot carbon storage and tree coverage/forest coverage area at different land uses (*Liu & Li, 2012*). In this study, forest type, administrative district, ring road, and history-related urban–rural gradients of the urban forests largely differed in carbon density, and approximate scaling-up showed fluctuations in total urban forest carbon storage from 521 to 575 thousand tons (Table 1). The different storage figures were an over simplified process; it is the product of forest area and corresponding carbon density in tree biomass and soils. Our findings highlight that, besides forest structure data, the comprehensive relationships among landscape metrics, urbanization gradients, and forest types could be included in a more precise evaluation framework. Several aspects should be considered.

First, as shown by the stepwise regression model between carbon storage parameters and landscape metrics, the possible landscape metrics suggested are LSI and LPI for carbon density and TA for total carbon storage estimation in tree biomass and soils (Table 4). With respect to biomass carbon storage, the inclusion of ENN_MN could increase the coefficient ($r^2$) from 0.65 (TA only) to 0.71 (TA and ENN_MN), whereas the soil carbon storage model showed that inclusion of AREA_MN increased $r^2$ from 0.97 to 0.99 (Table 4).

Second, urbanization intensity and forest types should be considered in the estimation model. As shown in Table 1, an approximation of total carbon sequestration based on forest types, administrative districts, and urban–rural gradients showed 1.2-fold differences in biomass and 1.1-fold variations in soil (Table 1). Urbanization-induced improvement of soil carbon has been reported. In our previous study, we found that the urbanization process (from rural to urban: fourth ring road to first ring road, 0-year region to 100-year region) led to the accumulation of SOC in Harbin (*Lv et al., 2016*), and the same findings were reported in Changchun, China (*Zhai et al., 2017*). The aggregation and shape complexity of patches decreased (in terms of LSI increased) along with the rural–urban gradients (Fig. 5), and the soil carbon storage density was negatively correlated with LSI (Fig. 6). This may possibly provide an explanation for SOC accumulation from the viewpoint of landscape. Small and shape-complex patches dispersed in the rural landscapes formed larger LSI and more easily exchanged substance and energy with the external environment than the big

and regular-shaped patches aggregated in the urban landscapes, including carbonaceous compounds, and may contribute to steady and higher carbon storage in central urban forests. In general, urbanization intensity differences and forest type differences are strongly associated with landscape variations in the forest.

Third, background urbanization conditions have been proved to be strongly associated with various ecological functions of urban forests, and their contributions to forest carbon sequestration need to be included. Building geography, e.g., street orientation (*Sanusi et al., 2016*); street canyon features, e.g., building height and distance to measured trees (*Coutts et al., 2016*; *Morakinyo et al., 2017*; *Rahman et al., 2017*);  land use configurations (building, street, green space, and water); and impervious surface percentage (*Zhang et al., 2017b*) have been proven to affect thermal regulation by trees, biodiversity conservation, and forest structural traits. In this paper, no such data are available, and future studies are required.

In the future, a well-matched, large dataset including all the above-mentioned parameters (independent of each other) would facilitate the derivation of a feasible method for scaling-up of city-level carbon storage density or total storage estimation. For example, landscape metrics in a fixed-sized area (e.g., 2 km around the field plot) (*Zhang et al., 2017b*), together with data on tree size and forest community features, soil carbon density and background road features, impervious surface features, as well as urban–rural gradients and forest types, will favor proper model construction for a proper scaling-up method via a statistical method such as the stepwise regression method used in this study (Table 4). This study has provided hints on the possible parameters for future consideration.

## CONCLUSIONS

The landscape of urban forests in Harbin was highly fragmented when compared with other local cities, and the fragmentation was different in different forest types, administrative districts, ring road- and urban history-related urban–rural gradients. The fragmentation of the landscape was strongly associated with carbon storage functions in trees and soils. LSI increased along with the urban–rural gradients, and its positive relationship with SOC indicates that LSI contributes to the underlying mechanism of urbanization-induced carbon accumulation in highly urbanized regions. The relationships between carbon storage and landscape metrics manifested possible ways to improve the urban forest carbon storage, such as improvement of the whole urban carbon storage by increasing the afforested area. An increase in the largest patch percentage and patch aggregation could increase the soil carbon storage per hectare. This study would help to promote carbon-oriented management practices for urban vegetation and exact evaluation of urban forest carbon sequestration by including landscape metrics and urbanization intensities.

## ACKNOWLEDGEMENTS

Thanks are due to Manli Ren and Zhongxue Pei for their help during field survey, and Professor Kaishan Song for his help while image processing.

### Funding

This study was supported financially by the One Hundred Talents Program in Chinese Academy of Sciences (Y3H1051001), Outstanding Youth Fund from Heilongjiang Province (JC201401), basic research fund for national universities from the Ministry of Education of China (2572014EA01; 2572017DG04), Key project from Chinese Academy of Sciences (KFZD-SW-302), NSFC project (31670699; 41730641). There was no additional external funding received for this study. The funders had no role in study design, data collection and analysis, decision to publish, or preparation of the manuscript.

### Grant Disclosures

The following grant information was disclosed by the authors:
One Hundred Talents Program in Chinese Academy of Sciences: Y3H1051001.
Outstanding Youth Fund from Heilongjiang Province: JC201401.
Ministry of Education of China: 2572014EA01, 2572017DG04.
Chinese Academy of Sciences: KFZD-SW-302.
NSFC project: 31670699, 41730641.

### Competing Interests

The authors declare there are no competing interests.

### Author Contributions

- Hailiang Lv conceived and designed the experiments, performed the experiments, analyzed the data, prepared figures and/or tables, authored or reviewed drafts of the paper, approved the final draft.
- Wenjie Wang conceived and designed the experiments, contributed reagents/materials/analysis tools, authored or reviewed drafts of the paper, approved the final draft.
- Xingyuan He conceived and designed the experiments, authored or reviewed drafts of the paper.
- Chenhui Wei, Lu Xiao, Bo Zhang and Wei Zhou performed the experiments, authored or reviewed drafts of the paper.

### Data Availability

    The raw data are provided as a Supplemental File.

### Supplemental Information

Supplemental information for this article can be found online at http://dx.doi.org/10.7717/peerj.5825#supplemental-information.

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
