# Peer review of "Association of urban forest landscape characteristics with biomass and soil carbon stocks in Harbin City, Northeastern China"

_PeerJ, doi:10.7717/peerj.5825_

## Round 0.1 · original submission · Major Revisions

Two anonymous reviewers have evaluated your manuscript. The main concern of your manuscript is that you have not written clearly enough how you have monitored carbon stocks in your study. The main motivation of proper methodology section is that readers can reproduce your methods. In the current form of manuscript readers cannot do that. In addition, reviewer #1 is asking why you think that administrative boundary will effect carbon storage.

Please, read carefully all recommendations and comments made by reviewers. And improve your manuscript according to those comments. When you have done that you can submit the revised manuscript.

Overall evaluation: Major revisions needed.

Reviewer 1 ·

Basic reporting

I stopped reading this manuscript at the methods section. This is a paper supposedly about soil carbon and tree carbon stocks in the urban forest of Harbin City.
Whilst I think this paper look potentially interesting - there is insufficient detail in the methodology to come to any conclusions about the work. There is no detail about how or even whether they soil sampled in order to estimate carbon stocks. There is no information at all about how they converted any tree information they collected in their field plots to carbon storage. Without this information I am not willing to go through the rest of the manuscript.

Experimental design

I stopped reading this manuscript at the methods section. This is a paper supposedly about soil carbon and tree carbon stocks in the urban forest of Harbin City.
Whilst I think this paper look potentially interesting - there is insufficient detail in the methodology to come to any conclusions about the work. There is no detail about how or even whether they soil sampled in order to estimate carbon stocks. There is no information at all about how they converted any tree information they collected in their field plots to carbon storage. Without this information I am not willing to go through the rest of the manuscript.
Other points - I do not understand why the authors think that administrative boundary will effect carbon storage - trees and soils do not conform to administrative boundaries.

Validity of the findings

Cannot assess this as method were insufficient.

Additional comments

I think potentially this could be worthy of publication BUT the methods section is too incomplete to assess this. See above comments. Remember the purpose of a methods section is to allow other to reproduce your methods. This would be impossible for your manuscript as it is. Whilst you describe the GIS approach used the field studies information is completely inadequate.
How did you sample your soils for carbon?
How did you analyse your soils?
What tree parameters did you measure in your field plots?
Did you identify tree species?
Did you use allometric equations to convert tree information to biomass and then carbon?

Reviewer 2 ·

Basic reporting

I would recommend the authors to refine the English word usage and style. There are many typos and grammar errors in the manuscript.

Some literature needs to be added to enhance the background and discussion of the manuscript.

Figures and tables need further improvement.

See attachment PDF for revision.

Experimental design

I believe the research fits the scope of PeerJ.

Research question defines well and fills an unknown knowledge gap.

Methods are robust, but some parts (especially the linear regression part) need further clarification.

Validity of the findings

Data is robust and findings are significant.

Additional comments

Please see the annotated pdf for revision.

Annotated reviews are not available for download in order to protect the identity of reviewers who chose to remain anonymous.

---

## Round 0.2 · Major Revisions

Reviewers have checked your manuscript again, and they said that your manuscript has improved, but it still needs some modifications. Read carefully all comments and suggestions made by reviewers, and modify your manuscript accordingly.

Reviewer 1 ·

Basic reporting

The English need improving in this manuscript as it is difficult to follow and incorrect words are used in places. e.g. L74 coped - this is not the correct word - do you mean coupled? This occurs throughout manuscript - in another place the authors say mean average - they mean the same thing.
The relevant references are there it is just that the clarity of the writing needs improving - the manuscript could certainly be shortened.
Although in general the figures and tables are well presented and described there are issues. see below.

Experimental design

I think the aims of this research are interesting but they are not well defined and I think there are problems with the independence of the way they have set up the scaling up work.
Description of the sampling in the methods is now much better.

Validity of the findings

Fig. 5 - where are the error bars on this figure? Presumably you used an average in each class? Were the statistics run on the site specific data or the averages?
Table 2 - need to provide area figs for each of the different classes. Also it is unclear why you get different storage figures when you estimate in different ways here. If you think that Forest type, Administrative district, ring road and history all effect carbon storage then they should be not be treated independently or used to calculate 4 different storage estimates. For example, presumably LF occurred in all administrative districts, and across the urban gradient. In addition in relation to this and other averages presented there are no errors associated with the averages provided.
I do not think that the discussion is well written and does not concisely bring together the results and is also in some places lacking references - e.g. see l292-304.
Table 4 - what is area mean? if you are using an area mean there is only one point and so how have you run a correlation - this in part would be solved if you had presented the d.f. or sample size in the analyses. Presumably the area mn is based on some kind of classification of the different patch area mean in different groups (e.g. LF)?

Additional comments

L123 - do you mean million not billion
Please try to report the data using the same number of decimal places.
L219 - presenting total storage in different districts is pretty meaningless to a global audience - there are no data provided on the area of trees in each district.
L395 - comparing the topsoil and aboveground carbon storage only beneath trees doesn't give any indication of the importance across an urban area of soil carbon storage relative to aboveground storage. Grassland covers the majority of the greenspace in an urban area and the carbon storage there is considerable greater than that aboveground.
l415-416 - there is no practical mention of how and where this afforestation might occur - it is not going to really be feasible in existing greyspaces and so generally it will occur in the existing grassland in the city.

Reviewer 2 ·

Basic reporting

English and literature cited are fine. Figs and tables need some improvement (see attachment).

Experimental design

Research fits with PeerJ scope. Research question defines well, and methods are robust.

Validity of the findings

Findings provide insights about urban forest fragmentation at Harbin.

Additional comments

After the major revision, the manuscript improves significantly. However, several minor revisions are needed to make this manuscript more completed (see attachment). The only thing I am concerned now is the accuracy assessment of image processing. Please follow my advice and improve it.

Annotated reviews are not available for download in order to protect the identity of reviewers who chose to remain anonymous.

---

## Round 0.3 · accepted · Accept

One anonymous reviewer has checked this revised version of your manuscript. The reviewer states that now this manuscript is suitable for publication.

Reviewer 1 ·

Basic reporting

Reporting int he manusucript is fine.

Experimental design

This is adequate.

Validity of the findings

Findings are valid.

Additional comments

L330 doesn't make sense.
Abstract results - no mention of soil just forest cover.
L384 - you have no evidence that tree planting in greenspace will increase soil carbon storage. You have not examined whether there are any differences in soil carbon between tree covered soils and soils in greenspaces covere with other vegetation.